# Yellow Twig (*Nauclea* *orientalis*) from Thailand: Strictosamide as the Key Alkaloid of This Plant Species

**DOI:** 10.3390/molecules27165176

**Published:** 2022-08-14

**Authors:** Weerasak Songoen, Julia Brunmair, Florian Traxler, Viktoria Chiara Wieser, Witthawat Phanchai, Wanchai Pluempanupat, Lothar Brecker, Johann Schinnerl

**Affiliations:** 1Department of Chemistry and Center of Excellence for Innovation in Chemistry, Special Research Unit for Advanced Magnetic Resonance, Faculty of Science, Kasetsart University, Bangkok 10900, Thailand; 2Department of Organic Chemistry, University of Vienna, Währinger Strasse 38, A-1090 Vienna, Austria; 3Department of Analytical Chemistry, University of Vienna, Währinger Strasse 38, A-1090 Vienna, Austria; 4Department of Botany and Biodiversity Research, University of Vienna, Rennweg 14, A-1030 Vienna, Austria; 5Department of Physics, Faculty of Science, Khon Kaen University, Khon Kaen 40002, Thailand

**Keywords:** *Nauclea orientalis*, Rubiaceae, monoterpene indole alkaloid, enzymatic deglucosylation, strictosamide

## Abstract

Comprehensive phytochemical examination from different perspectives using preparative and analytical chromatographic techniques combined with spectroscopic/spectrometric methods of the so-called “yellow twig” *Nauclea* *orientalis* (L.) L. (Rubiaceae) led to the identification of 13 tryptamine-derived (=monoterpene-indole) alkaloids. The identified alkaloids comprise strictosamide and four of its glucosidic derivatives, three oxindole derivatives, and five yellow-colored angustine-type aglycones. Qualitative and quantitative HPLC analyses showed the enrichment of strictosamide in all studied organs. Based on these results, we performed metabolomic analyses of monoterpene-indole alkaloids and made a ^1^H NMR in vitro monitoring of enzymatic deglucosylation of strictosamide. A comparison of the stability of strictosamide and its enantiomer vincoside lactam by theoretical calculations was also performed revealing a slightly higher stability of vincoside lactam. Additionally, we conducted two different anti-feedant assays of strictosamide using larvae of the polyphageous moth *Spodoptera* *littoralis* Boisduval. The obtained results indicate that generally two different biosynthetic pathways are most likely responsible for the overall alkaloid composition in this plant. Strictosamide is the key compound in the broader pathway and most likely the source of the identified angustine-type aglycones, which may contribute significantly to the yellow color of the wood. Its cross-organ accumulation makes it likely that strictosamide is not only important as a reservoir for the further biosynthesis, but also acts in the plants’ defense strategy.

## 1. Introduction

The genus *Nauclea* (Rubiaceae: Naucleaee) comprises 12 species [1] and occurs mainly in tropical Africa and Southeast Asia. The herein investigated species *Nauclea orientalis* (L.) L. is found from Sri Lanka to Southeast Asia and Northern Australia in various parts of Thailand and adjacent countries. The vernacular name “Yellow Twig” in Thailand or “Yellow Cheesewood” in Australia refers to its yellow wood after wounding. Leaves and bark of this plant species are mainly used in the local folk medicine as pain relievers after bites from animals, wounds, and abdominal pains [2,3]. Other ethnomedicinally important African *Nauclea* species are *N*. *latifolia* Sm., *N*. *officinalis* (Pierre ex Pit.) Merr. & Chun, *N*. *diderrichii* (De Wild.) Merr., and *N*. *pobeguinii* (Hua ex Pobég.) Merr. The known bioactivities from extracts of these plant species include antiplasmodial, antimicrobial, and analgesic effects [4,5].

From the phytochemical point of view, the hitherto studied *Nauclea* species are known for their accumulation of tryptamine-derived alkaloids, but only a few species have been studied so far. The afore-mentioned species accumulate strictosamide (**1**) together with many structurally related derivatives, most of them possessing a strictosamide core structure, e.g., [5,6,7]. *N*. *officinalis* oxindole-type indole alkaloids like nauclealomide A and some of its derivatives as well as (3*S*,7*R*)-javaniside were also reported [8,9,10].

From the species *N*. *orientalis* (Figure 1), several phytochemical studies have been undertaken so far, and all of them report tryptamine-derived alkaloids. Kanchanapoom et al. [11] studied leaves from an individual collected in the west of Thailand. He et al. [12] studied the stem of an individual collected in Laos, Sichaem et al. [3] investigated roots from the province Mahasarakham in Eastern Thailand, and Liu et al. [2] examined leaves and stems from Yunnan, China. All these studies revealed the predominance of strictosamide (**1**) or oxidized derivatives possessing the identical core structure as well as the co-occurrence of angustine-type aglycones in this plant species. 

Our hitherto accomplished studies on the occurrence of such alkaloids in neotropical *Palicourea* species (Rubiaceae) [13,14] (and references cited in there) together with the extraordinary yellow wood prompted us to study the alkaloidal composition of *N*. *orientalis*. Since previous studies reported the predominance of strictosamide (**1**) in this plant species, the following research questions appeared important to us: (i) Which alkaloids are accumulated in notable amounts and is there an observable organ-specific accumulation? (ii) Are there any biosynthetic connections between them and is strictosamide (**1**) a key intermediat? (iii) Is the pattern of alkaloids and/or predominance of (**1)** advantageous for the plant species, e.g., regarding herbivory?

To answer these questions, we performed several experiments including metabolomics, enzymatic in vitro degradation of strictosamide, and an antifeedant assays using the polyphageous moth *Spodoptera littoralis* Boisduval. The obtained results are discussed in chemical and biological contexts.

## 2. Results

The phytochemical investigation of leaf and stembark extracts of *N*. *orientalis* collected in the northeastern region of Thailand led to the identification of eight alkaloid glucosides and five aglycones. Alkaloid glucosides are namely strictosamide (**1**), 10-hydroxy strictosamide (**2**), 10-hydroxy vincoside lactam (**3**), 10-hydroxy-3-*epi*-pumiloside (**4**), and 3-*epi*-pumiloside (**5**), as well as the oxindole-type alkaloids nauclealomide B (**6**), nauclealomide C (**7**), and the oxindole derivative paratunamide C (**8**). Compounds **9**–**13** are the aglycones angustine (**9**), 3,14-dihydro angustine (**10**), 18,19-dihydroangustine (**11**), angustoline (**12**), and 3,14-dihydroangustoline (**13**). All of these aglycones are suspected to be derived from strictosamide (**1**) after deglucosylation. Structures of all 13 compounds (Figure 2) are unambiguously identified by 1D- and 2D-NMR experiments together with mass spectrometry, even if not all compounds were isolated to perfect purity. The spectroscopic and spectrometric data of compounds **1** to **13** are summarized in the Appendix A).

### 2.1. Structure Elucidation 

#### 2.1.1. Compounds **1**–**5**

Strictosamide (**1**) (HR-TOF-ESI-MS *m/z* 521.1896 [M+Na]^+^ (calcd. 521.1900 for C_26_H_30_N_2_O_8_Na^+^)) was isolated as the major compound from the leaf extract (Figure 2). The ^1^H NMR signals at δH 7.39 ppm (d, H-9), δH 7.33 ppm (d, H-12), δH 7.08 ppm (t, H-10), and δH 7.00 (t, H-11) indicated the signals of an indole ring. The secoiridoid glucoside moiety is assigned by the particular highly deshielded olefinic proton at δH 7.38 ppm (H-17) as well as by the ^2,3^*J*_H–C_ couplings between H-17 and C-15/C-16/C-21 and C-22. The characteristic proton signals at δH 5.66 ppm (dt, H-19), δH 5.37 ppm (dd, H-18a), and δH 5.32 ppm (dd, H-18b) revealed the terminal vinyl group. Additionally, the ^1^H NMR signals in the typical area between δH 3.0 and 4.0 ppm indicated proton position 2’ to 6’ in the glucoside moiety. The large coupling constant of an anomeric proton at δH 4.58 ppm (d, ^3^*J*_H-H_ = 7.9 Hz) assigned the presence of the β-glucoside unit. Furthermore, the carbon signal of C-21 (δC 98.0 ppm) implies an acetal carbon, which is an *O*-glycosidic bond in this position. NOEs between H-21 and H-1’ as well as ^3^*J*_H–C_ couplings between C-21 and H-1’ as well as C-1’ and H-21 in the HMBC spectrum show the β-glucoside to be bound in this position.

The relative configurations of positions 15, 20, and 21 were determined by the corresponding ^3^*J*_H-H_ coupling constants and the NOEs of the protons in these positions to the spatially close moieties. The resulting relative stereochemistry is in very good agreement with those of the corresponding biosynthetic iridoid precursor secologanin [15]. This is further supported by the detection of a cross peak between H-3 and H-20 in NOESY, which can be assigned to protons H-3 and H-20, which are located spatially on the same side. These data are supported by data reported earlier including the entire stereochemistry [16].

The NMR signal patterns of compounds **2** (HR-TOF-ESI-MS *m/z* 537.1856 [M+Na]^+^ (calcd. 537.1849 for C_26_H_30_N_2_O_9_Na^+^)) and **3** (HR-TOF-ESI-MS *m/z* 537.1845 [M+Na]^+^ (calcd 537.1849 for C_26_H_30_N_2_O_9_Na^+^)) were similar to those of strictosamide (**1**). However, these two compounds possess a hydroxyl group at C-10. This substitution was confirmed by the ^13^C NMR signal at δC 151.6 ppm and 151.4 ppm, respectively. For compound **2,** the stereochemistry at C-3 was the same *S*-configuration as in strictosamide (**1**). Hence, **2** was identified as 10-hydroxystrictosamide. For compound **3**, however, the stereochemistry at C-3 was inverted to the *R*-configuration, which indicated the presence of 10-hydroxyvincoside lactam (**3**). The inversion of position 3 was confirmed by NOEs and ^3^*J*_H-H_ coupling constants as well as by comparison of the spectroscopic data reported previously [16,17]. For further general investigation of the two enantiomeric forms, see Section 2.2.

For compound **4,** the 1D and 2D NMR spectroscopic data as well as mass spectrometric data (HR-TOF-ESI-MS *m/z* 551.1635 [M+Na]^+^ (calcd. 551.1642 for C_26_H_28_N_2_O_10_Na^+^)) imply the structure of 10-hydroxy-3-*epi*-pumiloside (**4**), as shown in Figure 2. Here, the quinolone ring is indicated by chemical shifts and multiplicities of ^1^H and ^13^C nuclei in position 2 as well as positions 6 to 13. Particularly, ^1^H NMR signals at δH 7.59 ppm (d, H-12), δH 7.58 ppm (s, H-9) and δH 7.08 ppm (dd, H-11), as well as the ^13^C NMR signal at δC 170.3 ppm (C-7), displayed the ABX aromatic ring system and carbonyl group of quinolone ring, respectively. In addition, ^2,3^*J*_H–C_ couplings in HMBC between H-9 and C-8/C-10 together with those between H-12 and C-8/C-10 indicated the substitution of a hydroxyl group at position 10. The characteristic signals of the secoiridoid and β-glucoside moiety were similar to compound **1**. 10-Hydroxy-3-*epi*-pumiloside (**4**) is described for the first time.

The NMR signals of 3-*epi*-pumiloside (**5**) (HR-TOF-ESI-MS *m/z* 513.1846 [M+H]^+^ (calcd. 513.1873 for C_26_H_29_N_2_O_9_^+^)) were almost identical to the signals of compound **4**. However, an unsubstituted at C-10 in aromatic ring is indicated by the four proton NMR signals at δH 8.28 ppm (dd, H-9), δH 7.43 ppm (t, H-10), δH 7.72 ppm (t, H-11), and δH 7.65 ppm (d, H-12) [18]. 

#### 2.1.2. Compounds **6**–**8**

Compounds **6**, **7,** and **8** were isolated from stembark of *N. orientalis* and identified as the monoterpene oxindole alkaloid derivatives nauclealomide B (**6**) (HR-TOF-ESI-MS *m/z* 553.1796 [M+Na]^+^ (calcd. 553.1798 for C_26_H_30_N_2_O_10_Na^+^)), nauclealomide C (**7**) (HR-TOF-ESI-MS *m/z* 553.1790 [M+Na]^+^ (calcd. 553.1798 for C_26_H_30_N_2_O_10_Na^+^)), and paratunamide C (**8**) (HR-TOF-ESI-MS *m/z* 569.1729 [M+Na]^+^ (calcd. 569.1747 for C_26_H_30_N_2_O_11_Na^+^)). The ^1^H NMR signals of the aromatic ring were in typical area between δH 6.8 and 7.9 ppm, and ^13^C NMR signals of the lactam group at δC 179.5 ppm (C-2) displayed the signals of oxindole ring system in compound **6**. Moreover, the downfield shifts of ^1^H NMR signal at δH 6.09 ppm (dd, H-3) as well as of ^13^C NMR signal at δC 76.3 ppm (C-7) and a ^3^*J*_H–C_ coupling between these two nuclei indicated the oxygen located between these two positions forming an ether bond linkage. The NMR spectra of compound **7** were very similar to compound **6**. However, the larger coupling constant of proton of position 3 in compound **6** (δH 6.09, dd, 9.5, 4.6) compared with H-3 in compound **7** (δH 5.76, dd, 3.9, 2.0) confirmed the stereochemistry of βH (3*S*) and αH (3*R*) for nauclealomide B (**6**) and nauclealomide C (**7**), respectively. The disappearance of the ^1^H NMR signal at position 3, concomitant with the appearance of a lactam group signal in ^13^C NMR (δC 173.6 ppm, C-3) in compound **8,** indicated the disconnection of ether bond between C-2 and C-3 and the oxidation of position 3. This ring opening leads to the structure of paratunamide C (**8**). The structure and relative stereochemistry of the three oxindole alkaloids **6** to **8** were confirmed by comparison of the spectroscopic and spectrometric data reported previously [9,17].

#### 2.1.3. Compounds **9**–**13**

Compound **9** to **13** were identified as the five indolopyridine alkaloids showed in Figure 2. The pyridine ring is indicated in ^1^H NMR by two singlets of the protons of positions 17 and 21 in typical region between δH 8.6 and 9.4 ppm. Angustine (**9**) (HR-TOF-ESI-MS m/z 314.1269 [M+H]^+^ (calcd. 314.1293 for C_20_H_16_N_3_O^+^)), 18,19-dihydroangustine (**11**) (HR-TOF-ESI-MS m/z 316.1439 [M+H]^+^ (calcd. 316.1450 for C_20_H_18_N_3_O^+^)), and angustoline (**12**) (HR-TOF-ESI-MS m/z 332.1393 [M+H]^+^ (calcd 332.1399 for C_20_H_18_N_3_O_2_^+^)) showed an additional highly deshielded olefinic proton signal of H-14 (δH 7.1 to 7.3 ppm). Compounds 3,14-dihydroangustine (**10**) (HR-TOF-ESI-MS m/z 315.1378 [M]^+^ (calcd. 315.1372 for C_20_H_17_N_3_O^+^)) and 3,14-dihydroangustoline (**13**) (HR-TOF-ESI-MS m/z 356.1359 [M+Na]^+^ (calcd. 356.1369 for C_20_H_19_N_3_O_2_Na^+^)) displayed the ^1^H NMR signal of position 3 at δH 5.09 and 5.00 ppm. Further NMR measurements indicated the difference of the functional group at positions 18 and 19. Their chemical structures were proven finally for all five compounds by comparison of the NMR data with those reported by [19]. 

### 2.2. DFT Caculations

Strictosamide (**1**, *α*H), and its enantiomer vincoside lactam (βH) differ in stereochemistry at C-3 position as shown in the 3D structures displayed in Figure 3. In order to get insights in the molecular stability of strictosamide (**1**) and vincoside lactam, we performed DFT calculations using Gaussian 09 software (*Rev. B.01*; Gaussian, Inc.: Wallingford, CT, USA) [20]. The lower total optimization energy of vincoside lactam (−1720.5249 a.u, −4.516377 × 10^6^ kJ mol^−1^) suggests its higher stability than strictosamide (−1720.5149 a.u, −4.516351 × 10^6^ kJ mol^−1^). The larger angle between protons H-3 and H-14 (*Φ* = 50.18°) as well as H-14 and H-15 (*Φ* = −66.91°) of vincoside lactam possess less strain energy between these protons (more staggered conformation). By contrast, the spatial proximity of H-3 and H-14 (*Φ* = −41.49°) as well as H-14 and H-15 (*Φ* = −24.39°) of strictosamide (**1**) affects the stability of its structure due to the more eclipsed conformation. 

### 2.3. Metabolomics Using LC-MS

For analyzing the metabolome, we created a library of 23 isolated indole alkaloids from *N*. *orientalis* and also different *Palicourea* species [21,22,23] (see Section 5.7). With this method, we detected compounds **1**, **2** (and/or **3**), **4,** and **5** as well as **12** in the crude methanolic leaf extract and compounds **2** (and/or **3**), **4**, **5**, **6**/**7**, **9**, **12,** and **13** in the stembark extract. The probable tryptophan-derived indole alkaloid 5α-carboxystrictosidine was also detected in the stembark extract (Appendix A). The stembark extract showed a higher alkaloid diversity, especially concerning the aglycones angustine (**9**), angustoline (**12**), and 3,14-dihydroangustoline (**13**). In turn, angustoline (**12**) was the only aglycon detectable in the leaf extract. Furthermore, the oxindole-alkaloids nauclealomide B/C (**6**/**7**) were only detectable in the stembark extract. The precursor of many monoterpene indole alkaloids strictosidine, and its derivatives strictosidinic acid, lyaloside, and lyalosidic acid were neither detectable in the leaf nor in the stembark extract. Furthermore, none of structurally more complex alstrostine derivatives were detectable in these samples.

### 2.4. Organ Specific Accumulation and Quantification of Strictosamide *(**1**)*

The quantification of strictosamide (**1**) by HPLC-UV-PDA in the dried powdered leaves, bark, wood (HPLC profiles are given in Appendix A), and fruits of *N. orientalis* was carried out using strictosamide (**1**) as external standard in the concentration range from 75 µg mL^−1^ to 1200 µg mL^−1^. The calculated calibration parameters were y = 13,590x − 54.496 (R^2^ = 0.9994). Strictosamide (**1**) was accumulated in the highest concentration in the wood (1.92% w/w), followed by bark (0.89%), leaves (0.82%), and fruits (0.16%).

### 2.5. Enzymatic Deglucosylation of Strictosamide *(***1***)*

In order to get insight into the downstream processing of **1** towards deglucolysed alkaloids like compounds **9**–**13**, purified strictosamide (**1**) was subjected to an enzyme-catalyzed deglucosylation using DMSO-*d*_6_/D_2_O (1:9) as solvent. This hydrolysis was accomplished using the archeal hyperthermostable β-glucosidase CelB from *Pyrococcus furiosus* [24,25] to enable a moderate reaction rate using a thermostable enzyme at ca 25 °C. Reaction monitoring was performed by in situ recording ^1^H-NMR spectra of the mixture (Figure 4) [26].

The disappearance of several indicative proton signals during biocatalyzed reaction progress revealed the hydrolysis of the glucoside moiety. Well visible are the decreasing ^1^H NMR signals of the terminal vinyl group at δH 5.56 ppm (H-19), δH 5.32 ppm (H-18a), and δH 5.29 ppm (H-18b). These chemical shifts differ slightly from those in Appendix A, due to different solvent systems used in structure determination and in situ monitoring, respectively. Moreover, the appearance of several ^1^H NMR signals indicated the formation and accumulation of free glucose as well as of an aglycone, which is here indicated as compound **1a**.

For further investigation of the accumulated aglycon **1a**, an upscaled deglucosylation reaction with subsequent chromatographic separation of the reaction product was accomplished. This afforded compound **1a** (HR-TOF-ESI-MS *m/z* 359.1362 [M+Na]^+^ (calcd. 359.1372 for C_20_H_20_N_2_O_3_Na^+^)) as a mixture of different isomeric structures. All these structures are based on two aldehyde groups at positions C-17 and C-21. These aldehydes are partly present in different tautomeric enol forms, in particular at position C-17. The mainly accumulated key isomer of this isolation was determined as (*Z*)-2-((2*S*,12b*S*,*Z*)-3-(hydroxymethylene)-4-oxo-1,2,3,4,6,7,12,12b-octahydroindolo [2,3-a]quinolizin-2-yl)but-2-enal (**1a**). The structure is shown in Figure 1 and in excellent agreement to aglycon recently predicted for this deglucosylation [27]. Spectroscopic and spectrometric data of **1a** are in Appendix A.

### 2.6. Gustatory and Feeding Experiments

In order to get notions about the importance of **1** as a defense chemical, we performed a comparative gustatory experiment and also a non-choice feeding assay using larvae of the polyphagous moth *Spodoptera littoralis* (Noctuidae). The gustatory experiment was performed following previously reported protocols [28,29] with slight modifications. For comparison, we used caffeine, aristolochic acid, and d-salicine, which are perceived as bitter by insects [29]. The obtained data from both assays suggests weak feeding deterrent properties of **1** similar to the phenolic compound d-salicine against *S*. *littoralis* (Table 1), whilst caffeine and aristolochic acid exhibited much stronger deterrent effects in comparison to the control (69%).

To figure out which concentration of strictosamide (**1**) prevents the uptake of food, we performed a non-choice feeding assay, which was accomplished after Sudžuković et al. with some modifications [30]. This experiment showed that the masses of the consumed spiked food pellets (0.8%: 71%; 0.4%: 71%, 0.2%: 75%) were not substantially different from that of the control (72%).

## 3. Discussion

Investigation of *N*. *orientalis* revealed strictosamide (**1**) as the most abundant alkaloid in all organs of this plant species. The other identified monoterpene indole alkaloid glycosides **2**–**5** are structurally closely related derivatives of **1**. In contrast, the oxindole alkaloids **6**–**8** are most likely derived via an alternative biosynthetic pathway starting from 2-oxo-tryptamine [9,31]. Such oxidation at C-2 may direct the condensation of the amine moiety and the secologanin to spiro-connected ring systems as known from javaniside and its derivatives [21,31]. Apart from these alkaloid glycosides, five aglycones (**9**–**13**), most likely resulting from a strictosamide core structure, were also isolated and identified (see Section 2.1.3.). The performed metabolic profiling together with the quantification of strictosamide in the plant organs point towards a distinct organ specific accumulation of strictosamide (**1**) in the various organs (see Section 2.3 and Section 2.4). The lack of strictosidine as well as of the further above-mentioned derivatives suggests a central role of strictosamide (**1**) for the alkaloidal composition in *N*. *orientalis*. The remarkable high abundance of **1** in the wood suggests an important role of this alkaloid for this plant species. Most probable, **1** also serves as plant defense chemical. Corresponding antifeedant experiments with *S*. *littoralis* larvae (see Section 2.6) indicate such effects although the results in the accomplished assays were not that clear.

The accomplished enzymatic degradation of strictosamide (**1**) (see Section 2.5) led to the purification of the angustine-type aglycone **1a** (Figure 1). The structural similarities of **1a** and the isolated angustine-type aglycones **9**–**13** suggest that **1** serves as a possible source for the identified aglycones herein, and the identified dialdehyde **1a** might be a possible precursor for the formation of a pyridine ring in the presence of amines or ammonium ions. For a more detailed consideration, see Section 3.1. However, the insertion of the nitrogen atom is suspected to proceed in a late stage of the biosynthetic pathway, leading to angustine (**9**) or angustoline (**12**) and structurally related compounds [32].

According to our observations in the field, the color of a cutting surface of the wood turns from light yellow to dark yellow over a period of one week (Appendix A). This change in color could be caused by the degradation of **1** in the wounded tissue leading to such dark yellow aglycones like angustine (**9**), 18,19-dihydroangustine (**11**), angustoline (**12**), and also (*Z*)-2-((2*S*,12b*S*,*Z*)-3-(hydroxymethylene)-4-oxo-1,2,3,4,6,7,12,12b-octahydroindolo [2,3-a]quinolizin-2-yl)but-2-enal (**1a**).

### 3.1. Possible Biosynthetic Pathways

In Figure 5, we summarize some possible biosynthetic pathways that can lead to an accumulation of the alkaloids isolated in the present work. Here we start from tryptamine and secologanin, which in the first step react in a Pictet Spengler reaction to strictosidine or to paratunamide C (**8**), respectively. It should be noted that no strictosidine could be detected in the screening of alkaloids by LC-MS as described in Section 2.3. However, since this compound is regarded as the key intermediate in the biosynthesis of strictosamide (**1**), we assume that strictosidine was converted quantitatively to strictosamide (**1**) and not accumulated in the examined *N. orientalis* individual. Strictosamide (**1**), on the other hand, represents the key intermediate for most of the accumulated and isolated alkaloids by accomplishing different biosynthetic pathways. It thus appears that in *N. orientalis* compound **1** is accumulated as the last intermediate in a joint biosynthetic pathway towards several alkaloids.

Interestingly, no vincoside can be detected in the LC-MS measurements either. Likewise, no hint to vincoside lactam is found in the isolation. Thus, with the stereochemistry of the products present in the early biosynthesis, there are no indications that the Pictet Spengler reaction does not proceed stereoselectively. Furthermore, there seems to be no racemization at C-3 at this stage, as described for example in [33] at lower pH. Consequently, in *N. orientalis* an epimerization to vincoside lactam, whose conformation is slightly more stable (see Section 2.2), appears to be kinetically hindered. Hence, in *N. orientalis* the situation is different from those reported for other plants in [34], where strictosidine is further metabolized and vincoside is metabolically inert.

Remarkably, the three isolated compounds **3**, **4,** and **5** show an epimerization at position C-3 compared to the corresponding configuration in compound **1**. The two angustine derivatives **10** and **13** also possess a chiral center at position 3. Recently published CD spectra of the two diastereomers (3*S*,19*S*)-**13** and (3*S*,19*R*)-**13** [10] are in good agreement with the optical rotations [α]_D_^20^ = −172.5° and [α]_D_^20^ = −301.4° reported for these two compounds [19]. The 3,14-dihydroangustoline (**13**) isolated in this work, however, shows an optical rotation of [α]_D_^25^ = −19.4°. This comparable significant much smaller value cannot be explained by a sole diastereomeric mixture at C-19. Likewise, a slow dehydrogenation to angustoline-type structures observed during longer storage (2 years) cannot be the main cause here. Hence, the large difference to the values reported in [19] makes an epimerization at C-3 in the pathway including deglucosylation (Figure 5, steps f and g) very likely.

Based on previous reports, e.g., [37], the hetero-aromatized pyridine moiety present in compounds **9**–**13** could be formed in later steps of the biosynthesis by reacting with ammonium ions to introduce an amino group. According to the reported indolopyridine alkaloid synthesis based on biological strategies, this type of alkaloid could be formed via an intermediate derived from deglucosylation of strictosamide (**1**) [32,37,38].

A similar deglucosylation of strictosidine is well known as a key step in the biosynthesis of many downstream products in different organisms. However, the aldehyde aglycone directly resulting from hydrolysis is reported to be very reactive and has not yet been isolated. In the root of *N. orientalis*, on the other hand, the two compounds naucleaorals A and B have previously been isolated [3]. These compounds represent isomeric forms of the aglycon, which derives from the hydrolysis of vincoside lactam. Only the double bond of the vinyl group is isomerized resulting in the two *Z*/*E* isomers. We, however, did not detect a comparable aglycone from strictosamide (**1**) in the examined *N. orientalis* individual, although both strictosamide (**1**) and the follow up products **9**–**13** were isolated.

To investigate this key step in strictosamide (**1**) to more detail, we catalyzed deglucosylation of **1** with a non-native glucosidase (CelB) and monitored the hydrolysis with in situ NMR. In this transformation, the deglucosylation as well as the accumulation of a direct subsequent product (**1a**) could be observed. No further uncontrolled subsequent reactions towards various follow up products were detected in notable amounts. The subsequent isolation and structure elucidation of **1a** illustrate its stability, although the two aldehydes expected from the hydrolysis are present in the structure, albeit in a partially enolized form (Figure 1). In addition, *Z*/*E* isomerization of the C-19/C-20 double bond was observed, comparable to those reported for naucleaorals A and B [3].

Thus, the direct derivatives of strictosamide (**1**) and vincoside lactam are less reactive than those of strictosidine and are also accumulated in plant parts of *N. orientalis* [3]. This could be an advantage in the biosynthesis of subsequent products, since by inducing respective enzymes, a reservoir of strictosamide (**1**) and vincoside lactam can be accessed in a more targeted manner. Purely chemically caused, uncontrollable side reactions of the resulting aldehydes thus lead to less substrate loss.

## 4. Conclusions

Strictosamide (**1**) is a key substance in *N*. *orientalis* in two respects. On the one hand, this monoterpene indole alkaloid is accumulated as the most abundant derivative in all investigated organs, while no accumulation of its supposed biosynthetic precursor strictosidine can be detected. Due to such high abundance in the aerial parts of the plant, strictosamide (**1**) plays an important role for the alkaloidal composition of *N. orientalis*. It is very likely that the plant benefits from its accumulation. This compound may also contribute to the defense system of the plant, even if no significant deterrent effect was observable in the performed experiments.

On the other hand, the majority of nine isolated alkaloids can be biosynthetically traced back to strictosamide (**1**), while only three oxindole derivatives can be traced back to an alternative biosynthetic pathway. The occurrence of such different pathways in *N. orientalis* advocates diverging biosynthetic pathways similar to *Palicourea luxurians* (Rusby) Borhidi [21] or also *Uncaria* species [39]. However, *N. orientalis* even produces a large chemical diversity of alkaloids in the areal plant parts from a reservoir of strictosamide (**1**) alone. The structural diversity is further enhanced by possible epimerization and the associated generation of enantiomers and diastereomers.

Some of these compounds in the plant tissue are angustine-type aglycones. An in vitro enzymatic degradation of strictosamide (**1**) yielded the intermittent dialdehyde **1a**, which has a striking stability. This is in contrast to the uncontrolled degradation of strictosidine during herbivorous attacks [40]. The dialdehyde **1a** thus likely plays a subordinate role in a chemical attack on herbivores but can itself form a reservoir from which bioactive specialized metabolites can be produced with a certain chemoselectivity.

It is noteworthy that the enrichment of these angustine-type aglycones may be responsible for the yellow-colored tissue after wounding. This visual impression has led to the naming of *N. orientalis* as “Yellow Twig” in Thailand or as “Yellow Cheesewood” in Australia. Not only because of that, we point towards directing more efforts into the study of wood chemistry in phytochemical investigations, which appears to be neglected when compared to those of other organs such as leaves.

## 5. Experimental

### 5.1. General Experimental Procedures

Analytical HPLC analyses were performed on Agilent 1100 series with UV-diode array detector and a Hypersil BDS-C18 column (250 × 4.6 mm, 5 μm particle size). An aqueous solution containing 10 mM ammonium acetate (A) and MeOH (B) was used as the eluent. The following elution system was applied: From 20–90% B within 15 min, from 90–100% B within 0.1 min, and 100% B was kept for 5.9 min, with a flow rate of 1.0 mL min^−1^. The injected concentration of the crude extracts was set to 10 mg mL^−1^ in pure methanol (MeOH). The wavelength of detection was set at 230 nm (reference WL 360 nm). Preparative LC was done by HPLC (Agilent 1200 series with UV-diode array detector) using a reversed phase Kaseisorb LC ODS 2000 column, 250 × 10.0 mm, 5 μm particle size, at a flow rate of 7.0 mL min^−1^ and an injection volume of 100 µL. The concentration of the injected crude extract was set at 68.0 mg mL^−1^ in pure MeOH, and water (A) and MeOH (B) were used as eluents. The following gradient was applied: From 5–50% B within 10 min, from 50–70% B within 10 min, and kept at 70% B for 10 min, then from 70–95% B within 7 min. The wavelength of detection was set at 254 nm.

Medium-pressure liquid chromatography (MPLC) separations were obtained over a silica gel 60 column (40–63 µm particle size). Mixtures of petroleum ether (PE), ethyl acetate (EtOAc), and methanol (MeOH) were used as eluents. For preparative TLC (pTLC), silica gel F_254_ plates with 0.5 mm thickness (Merck) were used. For analytical TLC silica gel 60 F_254_ plates with 0.2 mm thickness (Merck) were used. All TLC plates were developed in various solvent mixtures consisting of PE, CHCl_3_, EtOAc, and MeOH. The purification based on size exclusion chromatography (SEC) was done by using Sephadex LH-20 (GE Healthcare) eluted with MeOH.

NMR spectra were recorded on a Bruker AVIII 600 spectrometer at 600 MHz (^1^H) and 150 MHz (^13^C), and Varian Unity Inova 400 spectrometer at 400 MHz (^1^H) and 100 MHz (^13^C). In situ ^1^H NMR measurements were recorded on a Bruker AVIII at 400 MHz (^1^H). The appropriate deuterated solvents CD_3_OD or DMSO-*d*_6_ were used to dissolve the isolated compounds (amounts 1–5 mg) in 0.6 mL each. Spectra were analyzed with MestReNova version 14.1.2, Santiago de Compostela, Spain. Chemical shifts (*δ*) for ^1^H NMR and ^13^C NMR are given in parts per million (ppm). For ^1^H NMR, the relative residual of non-deuterated solvent signals was referenced to CHD_2_OD (δ_H_ = 3.31 ppm) and DMSO-*d*_5_ (δ_H_ = 2.50 ppm), and for ^13^C NMR solvent signals were referenced to CD_3_OD, (δ_C_ = 49.0 ppm) and DMSO-*d*_6_ (δ_C_ = 39.5 ppm). The multiplicities of CH_3_, CH_2_, CH, and Cq are indicated by quartet (q), triplet (t), doublet (d), and singlet (s), respectively.

High-resolution mass (HRMS) spectra were recorded on a maXis UHR ESI-Qq-TOF mass spectrometer (Bruker Daltonics, Bremen, Germany). Each sample was dissolved and further diluted in acetonitrile/MeOH/H_2_O in the ratio of 99:99:2 (*v/v/v*) and directly infused into the ESI source with a syringe pump. The ESI ion source was operated as follows: capillary voltage: 4.0–4.5 kV, nebulizer: 0.4 bar (N_2_), dry gas flow: 4 L min^−1^ (N_2_), and dry temperature 180 °C. The ranges of *m/z* from 50–1900 in both positive and negative ion mode were recorded. The sum formulae of the detected ions were determined using Bruker Compass Data Analysis 4.1 based on the mass accuracy (Δ*m/z* ≤ 5 ppm) and isotopic pattern matching (SmartFormula algorithm). 

The optical rotation was measured by the sodium D line with a 100 mm of path length cell on a Perkin Elmer Automatic Polarimeter 341 (Perkin Elmer Austria) at a concentration of 0.95 mg mL^−1^ in MeOH.

For LC-MS analysis of the plant extracts, a Q Exactive HF (Thermo Fisher Scientific) mass spectrometer coupled to a Vanquish UHPLC System (Thermo Fisher Scientific) was employed. Chromatographic separation was achieved on a Kinetex XB-C18 column (100 Å, 2.6 µm, 150 × 2.1 mm, Phenomenex Inc.). Mobile phase A consisted of water with 0.2% formic acid and mobile phase B consisted of methanol (all VWR chemicals, Vienna, AT) with 0.2% formic acid. The following gradient program was run: 1–5% B in 0.3 min and then 5–40% B from 0.3–5 min, followed by 80% B from 5–6.9 min and a column equilibration phase at 1% B for 2 min, giving a total run time of 9 min. The flow rate was 0.5 mL min^−1^, the column temperature was 40 °C, the injection volume was 10 µL, and the injection peak was found at RT = 0.5 min. All samples were analyzed in technical replicates and every 5th injection was a blank (LC-MS grade water) injection. An untargeted mass spectrometric approach was applied. Electrospray ionization was performed in positive and negative ionization mode. MS scan range was *m/z* 100–1000, and the resolution was set to 60,000 (at *m/z* 200). Source parameters were the following: sheath gas flow rate of 40, auxiliary gas flow rate 2, and auxiliary gas heater temperature of 300 °C for both ionization modes; a spray voltage of 3.20 kV and a capillary temperature of 250 °C for positive mode, and a spray voltage of 2.80 KV and a capillary temperature of 320 °C for negative mode. The instrument was controlled using Xcalibur software (Thermo Fisher Scientific) and Q Exactive HF tune (Thermo Fisher Scientific).

### 5.2. Plant Material

Leaves and stems of *N. orientalis* were collected from Udon Thani, (17°06′58.3″ N, 102°35′33.2″ E), Thailand in May 2018. A voucher specimen (BK No. 071002) was deposited at the Bangkok Herbarium, Plant Varieties Protection Office, Department of Agriculture, Bangkok, Thailand.

### 5.3. Extraction and Isolation

#### 5.3.1. Isolation from Leaves

The air-dried ground leaves of *N. orientalis* (2.4 kg) were sequentially extracted (3 × 7 d) with PE, CH_2_Cl_2_, EtOAc, and MeOH. Each crude extract was filtered and evaporated under reduced pressure. The methanolic extract (162.6 g) was subsequently partitioned between distilled water, 5% EtOAc in PE, CHCl_3_ (3.0 g), and EtOAc (14.7 g). The chloroform fraction (3.0 g) was initially chromatographed by medium-pressure liquid chromatography (MPLC) over silica gel. The elution system started with PE, EtOAc, and MeOH with increasing polarity, which afforded 32 subfractions named NO-M01-F1 to F32.

The combined fractions of NO-M01-F30 and NO-M01-F31 (48.2 mg) were chromatographed over Sephadex LH20 eluted with MeOH, yielding 4.6 mg of impure **4**. The fraction impure **4** was further purified by preparative TLC eluted with 50% of MeOH in CHCl_3_. This step afforded 1.9 mg of **4**. Separation of fraction NO-M01-F24 (145.3 mg) by Sephadex LH20/MeOH afforded 25.3 mg of **1**. From fraction NO-M01-F25, 64.7 mg was subjected to SEC over Sephadex LH20/MeOH. This step furnished 11.4 mg of **2**. From fraction NO-M01-F26 (68.6 mg) 1.1 mg of **3** was obtained by size exclusion chromatography over Sephadex LH20 eluted with MeOH.

#### 5.3.2. Isolation from Stembark

Air-dried ground stembark of *N. orientalis* (2.4 kg) was subsequently extracted with PE, CH_2_Cl_2_, EtOAc, and MeOH, respectively (3 × 7 d). All extracts were filtered and evaporated under reduced pressure in a rotary evaporator. The EtOAc extract (7.1 g) was chromatographed over silica gel column chromatography gradually eluted from 60% EtOAc in PE to 60% MeOH in EtOAc. The separation afforded 10 subfractions (NO-EC01-F01 to F10). The fraction NO-EC01-F10 (750 mg) was chromatographed over Sephadex LH20/MeOH. This step afforded six fractions named NO-ES01-F01 to 06. The 117.1 mg of NO-ES01-F03 was subjected to MPLC over silica gel eluted gradually from 30% EtOAc in PE to 80% MeOH. This afforded 15.5 mg of impure **5**. Purification of **5** by MPLC yielded 5.1 mg.

The fraction NO-EC01-F09 (744 mg) was fractioned over silica gel column chromatography eluted isoratic with 10% MeOH in EtOAc. This step yielded seven fractions named NO-EC02-F01 to F07. The selected fraction NO-EC02-F05 (102.0 mg) was further chromatographed using preparative HPLC. This separation afforded 1.5 mg of **6**, 2.6 mg of **7** and 1.5 mg of **8**.

The CH_2_Cl_2_ extract (7.6 g) was fractionated over silica gel column chromatography gradually eluted with mixtures consisting of PE, EtOAc, and MeOH with increasing polarity. This separation step gave 17 sub-fractions named NO-DC01-F01 to F17. Fraction NO-DC01-F13 (168.2 mg was further purified by Sephadex LH20/MeOH. This furnished 2.3 mg of **9**, 90.7 mg of impure **10**, and 10.4 mg of impure **11**. Compound **10** (1.2 mg) was purified by Sephadex LH20/MeOH, while compound **11** (2.1 mg) was obtained from pTLC elution with 5% MeOH in EtOAc. The separation of fraction NO-DC01-F15 (451.3 mg) by Sephadex LH20/MeOH afforded 8.5 mg of **12** and 14.8 mg of **13** in a 1: 2 diastereomeric ratio.

#### 5.3.3. Isolation from Wood

Air-dried ground wood of *N. orientalis* (400 g) was extracted with MeOH (3 × 7 d). The MeOH extract (12.2 g) was filtered and evaporated under reduced pressure in a rotary evaporator and subsequently partitioned between distilled water and *n*-butanol. This afforded 4.9 g of crude *n*-butanolic extract. The *n*-butanol phase was further chromatographed by MPLC over silica gel using a steps-wise gradient elution started with 60% EtOAc in PE to 80% MeOH in EtOAc. This step afforded 188 mg of impure strictosamide (**1**). Final purification over Sephadex LH20/MeOH yielded 95.6 mg of strictosamide (**1**).

### 5.4. Theoretical Calculation

The calculations of optimization energy and frequency of single molecule were accomplished by using Gaussian 09 software (Rev. *B.01*; Gaussian, Inc.: Wallingford, CT, USA) [20]. The DFT calculations were used to observe an energy minima and vibration mode of single molecule. The DFT method was carried out by hybrid function Becke−3−Lee−Yang−Parr (B3LYP) and double-ζ polarized basis set with six d-type Cartesian−Gaussian polarization functions (6-31G(d,p)).

### 5.5. Enzymatic Deglucosylation

The enzymatic deglucosylation of strictosamide (**1**; 0.14 mmol) was carried out in 10% DMSO in phosphate buffer (20 mM, pH 6.0) at RT for 6 d. The β-glucosidase CelB (50 U final concentration) [24,25] was used for this hydrolysis. The final concentration of strictosamide (**1**) was 3.33 mM. After performing the reaction, the mixture was evaporated and purified by CC over Sephadex LH20 eluted with MeOH. This step afforded 10 fractions named SE02-F01 to F10. Fraction SE08 was further partitioned between distilled water and CHCl_3_. The CHCl_3_ residue (53.7 mg) was further purified by pTLC eluted with 25% of EtOAc in PE. This afforded 5.0 mg of **1a**. For further investigation of compound **1a**, the CelB catalyzed hydrolyses of strictosamide (**1**) was repeated in larger scale in non-deuterated solvent (10% DMSO/H_2_O (pH 6.0) 1:9). After a reaction time of 6 d, the reaction mixture was chromatographically separated and afforded compound **1a** as a mixture of different isomeric forms. This enzymatic deglucosylation was also performed for an in situ ^1^H NMR monitoring following the work of Brecker and Ribbons [26]. Therefore, the reaction was carried out in an NMR tube in 10% DMSO-*d*_6_ in phosphate buffer (20 mM, pD 6.04) in D_2_O at RT. ^1^H NMR spectra were recorded in regular intervals.

### 5.6. Quantification of Strictosamide (***3***)

The quantification of **1** was carried out by using purified strictosamide (**1**) as an external standard. The stock solution of strictosamide (**1**) (1.2 mg mL^−1^) was prepared in MeOH. From this stock solution, a dilution series comprising five standards in the concentration of 75, 150, 300, 600, and 1200 µg mL^−1^ was prepared. Each standard solution was analyzed by HPLC three times. The calibration curve was obtained by plotting between the mean of the peak areas versus concentration of strictosamide (**1**). For quantification, dried leaves, bark, and wood of *N. orientalis* were ground and sieved (0.20 mm) prior to extraction. The sieved residues were accurately weighed (100.1 mg) and extracted by MeOH (3 × 1.5 mL) under sonication for 15 min. The MeOH extract was centrifuged at 14,000 rpm for 20 min. Finally, the supernatants were pooled, the solvent evaporated, and the samples adjusted to 1.0 mg mL^−1^ in pure MeOH for HPLC analysis.

### 5.7. Screening of Alkaloids by LC-MS

The LC-MS Library composed of the isolated compounds **1**–**13** as well as the previously isolated indole alkaloids strictosidine and its derivatives strictosidinic acid, lyaloside, and lyalosidic acid were established. The compounds **1**–**7**, **9**, **10,** and **13** were isolated from *N. orientalis*. The other alkaloids 5α-carboxystrictosidine, deoxostrictosamide, javaniside, lyaloside, lyalosidic acid, strictosidine, strictosidine acid, vincosamide, javaniside, and the structurally more complex derivative alstrostine A were isolated from *Palicourea* species during previous studies [21,22,23]. The above-mentioned compounds and the crude methanolic leaf and stembark extracts of *N*. *orientalis* samples in the concentrations of 100, 1.0, and 0.1 ng mL^−1^ in methanol were prepared and injected. The analyses of the obtained chromatograms were performed under the parameters given in Section 5.1.

### 5.8. Antifeedant Experiments

#### 5.8.1. Gustatory Experiment

Briefly, each compound was distributed evenly on weighed leaf discs from commercial lettuce (*Lactuca sativa* L.). For compound **1** we used a concentration similar to its concentration in the leaves of *N*. *orientalis*. The leaf discs were distributed in 2, 4, 6, 8, 10, and 12 o’clock mode in the Petri dishes, the solutions applied accordingly, and one caterpillar at the third larval stage was placed in the center. The leaf disc at position 10 was the control treated with the solvent mixture and at position 12 o’clock was an untreated leaf disc. After 24 h the leaf discs were weighed again, and the differences were calculated. We used the differences as a measure of attractiveness and compared them to the reference chemicals used. The leaf discs treated with solvent were used for all the calculations. No differences were observable between the leaf discs treated with the solvent mixture and the untreated leaf discs. Stock solutions of **1**, caffeine, aristolochic acid, and d-salicine (5.0 mg mL^−1^ each) in acetone/water (90:10) were prepared, and 10, 20, or 30 µL of each solution were distributed evenly on each leaf disc (13 mm in diameter, stamped out from a common lettuce) to reach different amounts of substances on the discs (Table 1). The chosen amount of **1** and the three other compounds were mimicking the concentration of **1** present in the leaves (see Section 2.4). After evaporation of the solvent, each leaf disc was weighed. Humidified filter paper was put into a Petri dish (90 mm in diameter) and four of these leaf discs—one for each compound—were placed in positions 2, 4, 6, and 8 o’clock. In the center of each Petri dish a caterpillar of *S*. *littoralis* in the third larval stage was placed, which had been starving for 16 h. The caterpillars were allowed to consume the leaf discs for 24 h with a light/darkness/light regime of 8/8/8 at 24 °C. After this time the leaf discs were weighed again, and the mass differences were used as a measure for the attractiveness of the compounds. The calculated differences were corrected by the loss of water. To assess the average loss of water, we treated 18 leaf discs in the same manner and evaluated their mass losses after 24 h under the same conditions as described above.

#### 5.8.2. Non-Choice Feeding Assay

The non-choice feeding assay was performed in triplicate with slight modifications according to Srivastava and Proksch [28]. Briefly, 184 mg of freeze-dried food powder containing ground white beans, yeast, ascorbic acid, and ethyl *para*-hydroxy benzoate as a preservative was spiked with 0.8, 0.4, and 0.2% of **1**, also mimicking its concentration in the leaves. After evaporation of the solvent (MeOH, 16 h), 363 µL of an aqueous solution consisting of 352 µL ddH_2_O, 11.5 µL vitamin solution and 187.5 µg of the antibiotic chloramphenicol was added, and the powder was solidified by adding 0.6 mL of a warm (50 °C) agar-agar solution (5 g in 140 mL water). These food pellets were transferred into Petri dishes, and a third instar larvae of *S. littoralis* was placed on each food pellet. The Petri dishes were kept in an incubator at 26 °C and 90% humidity in darkness for 72 h. Afterwards, the masses of the remaining food pellet were assessed. Unspiked food pellets served as controls. All the calculated values were corrected by the loss of water.

## Data Availability

Data can be obtained from the corresponding authors.

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
