# Peer review of "Yellow Twig (Nauclea orientalis) from Thailand: Strictosamide as the Key Alkaloid of This Plant Species"

_molecules, 2022, doi:10.3390/molecules27165176_

Round 1
Reviewer 1 Report
Dear Songoen et al.,
Songoen et al. reported the isolation of 13 compounds from different part of plant Nauclea orientalis collected at Thailand. In addition, compound 3 showed slight anti-feeding activity against worm.
I enjoy reading the manuscript, although this is not groundbreaking discovery, but a solid natural product chemistry work.
No further suggestion from me.
Author Response
Thank you very much.
Reviewer 2 Report
The authors undertake a comprehensive phytochemical examination of the so-called “yellow twig” Nauclea orientalis (L.) L. (Rubiaceae) and isolated and identified 13 tryptamine-derived alkaloids, including strictosamide and four derivatives, three oxindole derivatives and five angustine-type aglycones. Qualitative and quantitative HPLC analyses showed strictosamide (1) was a key alkaloid of this medicinal plant. Further 1H NMR in situ enzymatic deglucosylation of strictosamide (1) as well as the large-scale preparation of its aglycone (1a) were performed. The stability of strictosamide and its enantiomer vincoside lactam was compared through theoretical calculations. Two biosynthetic pathways for strictosamide (1) typed alkaloid and oxindole typed alkaloid were hypothesized, and their relationship with angustine-type aglycones were proposed. Considering the enrichment of strictosamide (1) and its potential as a defense chemical produced by this plant, the anti-feedant activity of strictosamide (1) was assayed.
The author collected all of these information from different perspectives and try to tell readers a story focusing mainly on strictosamide, its isolation from N. orientalis, content, stability, biosynthesis, and function as well as its possible relationship with angustine-type aglycones from biosynthetic viewpoint. These results will broaden our knowledge on key strictosamide and the possible reason for “yellow color” formation for Nauclea orientalis, and provide the important basis for future biosynthetic and biological research.
1. There are some grammatical problems in the manuscripts which are highlighted in the revised manuscript, some sentences even confused me and they need to be intensively polished.
2. In abstract, the biosynthetic pathway is only hypothesized, it can not be confirmed by these obtained results.
3. What conclusion can you draw from the stability comparison of strictosamide and its enantiomer vincoside lactam? Please add it in the abstract.
4. How about the novelty of compounds 1-13? Are they new compound, or known ones, or first isolated from this medicinal plant? Please make it clear.
5. You just describe the relative configuration of strictosamide (1). How do you determine the absolute configuration of H-3 is S, not R? Using ECD calculation?
6. 2.1.1 In compound 5, since there was no symmetric structure, the aromatic protons were not attributed to AA'BB' signals.
7. How to explain this change for H-3 in Fig.4? Why a sharp singlet of H-3 at about 5.0 ppm become a broad doublet? Are Keto-enol tautomeric forms in the ratio of 1:1?
8. What is the peak at about 2.1 ppm in Fig.4? Why does it not exist at 0h and appear after 6h?
9. β-glucosidase is also an organic matter, does it have some interruption to NMR signals in in situ hydrolysis experiment as shown in Fig.4?
10. How about the color of compound 1a? Does it take on a dark yellow color similar with angustine-type aglycones 9, 11, and 12, and also responsible for the wood color change?
11. Page 13, “In this pathway compound 1a is probable intermediate within step f, which is however not accumulated due to its reactivity”. However, from your experiment, 1a possessing two aldehyde groups seems stable under room temperature.
12. Other minor mistakes were highlighted in the revised manuscript.

Author Response
- There are some grammatical problems in the manuscripts which are highlighted in the revised manuscript, some sentences even confused me and they need to be intensively polished.
Thank you very much for highlighting these sentences. The whole manuscript was corrected.
- In abstract, the biosynthetic pathway is only hypothesized, it can not be confirmed by these obtained results.
We modified this part accordingly.
- What conclusion can you draw from the stability comparison of strictosamide and its enantiomer vincoside lactam? Please add it in the abstract.
This was added to the abstract.
- How about the novelty of compounds 1-13? Are they new compound, or known ones, or first isolated from this medicinal plant? Please make it clear.
The identified compounds, except compound 4, are known. Compound 4 is not yet published in SciFinder.
- You just describe the relative configuration of strictosamide (1). How do you determine the absolute configuration of H-3 is S, not R? Using ECD calculation?
Our data are in agreements with previous reports in which the stereochemistry of 1 was accurately determined.
- 1.1 In compound 5, since there was no symmetric structure, the aromatic protons were not attributed to AA'BB' signals.
This part has been corrected.
- How to explain this change for H-3 in Fig.4? Why a sharp singlet of H-3 at about 5.0 ppm become a broad doublet? Are Keto-enol tautomeric forms in the ratio of 1:1?
The signal, at about 5 ppm disappeared and new signals appeared due to the formation of further compounds during this reaction.
- What is the peak at about 2.1 ppm in Fig.4? Why does it not exist at 0h and appear after 6h? This signal belongs to acetic acid from the buffer solution and is not present at t=0 because no buffer was present at this time.
- β-glucosidase is also an organic matter, does it have some interruption to NMR signals in in situ hydrolysis experiment as shown in Fig.4?
This enzyme is only present in tiny conc. and has shown very small and broad signals as proteins normally do.
- How about the color of compound 1a? Does it take on a dark yellow color similar with angustine-type aglycones 9, 11, and 12, and also responsible for the wood color change?
The color of 1a was deep yellow, similar to 9, 11 and 12. Therefore, we assume the changing color of the wood is directly linked with the degradation of strictosamide. In leaves this color is not visible due to the chlorophylls present in these photosynthetic organs.
- Page 13, “In this pathway compound 1a is probable intermediate within step f, which is however not accumulated due to its reactivity”. However, from your experiment, 1a possessing two aldehyde groups seems stable under room temperature.
This compound was relatively stable under these artificial conditions but it might be not that stable under natural conditions in plant cells.
- Other minor mistakes were highlighted in the revised manuscript.
The manuscript has been revised and we corrected the mistakes.
Reviewer 3 Report
The manuscript can be published in Molecules after the authors add the methods from Supplementary materials (S1.0 – S1.4) in the manuscript. This will help readers to understand the paper easily.Author Response
The manuscript can be published in Molecules after the authors add the methods from Supplementary materials (S1.0 – S1.4) in the manuscript. This will help readers to understand the paper easily.
These parts were shifted to the manuscript.
Reviewer 4 Report
Dear Authors,
the work entitle "Yellow Twig (Nauclea orientalis) from Thailand: Strictosamide as the key alkaloid of this plant species" was well designed and conducted. This work suggests a very interesting correlation on the biosynthesis between the metabolites, and presented as the chemical characterization of some of them. In my opinion, the work is quite good, with just few points to be clearer.
1- please, insert or clarify the chemical isolation of the epimers compounds 2/3; and 6/7. I didn't find how compound 3 was obtained.
2-The methodology of isolation is nor clear to me. It is a bit confusing about the use of MPLC or prep HPLC. Column used?
3-Why didn't u checked the optical rotation of those metabolites? Some 2 D NMR data should be important to characterize those metabolites, but I just found the 1HNMR in the supplementary data. Once MS spectrum is the same, I was expecting some optical rotation, CD analises or 2D complementary information.
Author Response
- please, insert or clarify the chemical isolation of the epimers compounds 2/3; and 6/7. I didn't find how compound 3 was obtained.
This part has been corrected.
2-The methodology of isolation is nor clear to me. It is a bit confusing about the use of MPLC or prep HPLC. Column used?
In this work we used two HPLC instruments, Agilent 1100 for analytical and Agilent 1200 for preparative purposes, Additionally, we used MPLC for performing some separation steps. This has been clarified in the manuscript.
3-Why didn't u checked the optical rotation of those metabolites? Some 2 D NMR data should be important to characterize those metabolites, but I just found the 1HNMR in the supplementary data. Once MS spectrum is the same, I was expecting some optical rotation, CD analises or 2D complementary information.
Since all the compounds, except compound 4, are known, we only show the proton and 13C spectra in the supplement file. But from all isolated compounds a complete set of 1D and 2D NMR spectra together with the MS spectra were recorded and the interpretation was done manually for each compound. Optical rotations were measured were necessary (e.g. compound 13).
Round 2
Reviewer 2 Report
The authors make a major revision according to reviewer's suggestion, and the quality of the manuscript is improved a lot. I am satisfied with these corrections. The sentence "We are also grateful to the four reviewers for their time and effort" in the Acknowledgement is suggested to be deleted.
Reviewer 3 Report
The authors were added the experimental procedures in Methods as suggested. The manuscript is recommended for publication in Molecules.
This manuscript is a resubmission of an earlier submission. The following is a list of the peer review reports and author responses from that submission.
Round 1
Reviewer 1 Report
The manuscript molecules-1419315 by Songoen et al. is focused on the study of eight tryptamine-derived alkaloid glucosides, and quantification of strictosamide using HPLC- DAD. Their molecular structure was indentificated by 1H NMR and 13C NMR.
After sample preparation, a screening of plant extracts ware analyzed by LC-MS ( UHPLC) analysis.
Although the novelty is not great from an analytical methodologicaly aspect, the accurate identification of the structures of the compounds studied was a meticulous job.
The manuscript is presented as a specific phytochemical study of leaf and stembark extracts of a particular plant, Nauclea orientalis.
The work appears to be well organized, although this manuscript is found to have a slight novelty, in my opinion, it does deserve to be published in Molecules.
Reviewer 2 Report
Comments:
The manuscript reports a phytochemical investigation on Nauclea orientalis, following by the quantitative analysis of strictosamide (3) in presence of each plant tissues, the removal of glucose of 3 by a non-native β-glucosidase, crude investigation of alkaloids by LCMS and antifeedant activity of 3 on the larvae of cotton leafworm. Initially negative impression comes from the abstract. The abstract does not state the significance of the study, if there have one. The study contents are more likely to be combined roughly. This reviewer would not endorse the conclusion that the importance of strictosamide (3) in Nauclea orientalis supported by the present study results.
In addition, all natural products shown in this manuscript have been reported in previous study. Low purity of compound 1, 2, 3a, 11 should be noted. This reviewer concerns that the low quality of reported natural products would not meet the publishing request of Molecules.
This reviewer concerns about the perfunctory determination of relative configuration of compound 1 based upon the statement of “The relative configurations of positions 15, 20 and 21 were deduced according to the known biosynthesis of secologanin”.
In figure 5, this reviewer concerns the biosynthetic pathway of 6, 7, 8. Compound 8 is more like the precursor of 6 and 7.
Reviewer 3 Report
This manuscript reported the isolation of 13 compounds from different part of plant Nauclea orientalis collected at Thailand. In addition, compound 3 providing a hint that it may serve as feeding deterrent against worm.
1) MHz of NMR should be mentioned in the NMR spectra found in supplementary materials and NMR table.
2) The novelty should be clearly mentioned in the abstract. Structural discussion for known compounds should be omitted.
3) The justification for determining the molecular stability of 3 and 3b is not strong. Does 3b present in the crude extract?
4) DFT calculation software should be cited.
5) Paragraph for enzymatic deglucosylation of 3 is confusing, the purpose and the question trying to attempt are not meaningful.
6) Only compound 3 was carried out for biological evaluation, why?